# Microbiological and Clinical Findings of SARS-CoV-2 Infection after 2 Years of Pandemic: From Lung to Gut Microbiota

**DOI:** 10.3390/diagnostics12092143

**Published:** 2022-09-02

**Authors:** Alessandro Russo, Francesca Serapide, Angela Quirino, Maria Grazia Tarsitano, Nadia Marascio, Riccardo Serraino, Salvatore Rotundo, Giovanni Matera, Enrico Maria Trecarichi, Carlo Torti

**Affiliations:** 1Infectious and Tropical Disease Unit, Department of Medical and Surgical Sciences, “Magna Graecia” University of Catanzaro, 88100 Catanzaro, Italy; 2Clinical Microbiology Unit, Department of Health Sciences, “Magna Graecia” University of Catanzaro, 88100 Catanzaro, Italy; 3Research Center for the Prevention and Treatment of Metabolic Diseases, “Magna Graecia” University of Catanzaro, 88100 Catanzaro, Italy

**Keywords:** SARS-CoV-2, COVID-19, gut microbiota, diagnostic test, extrapulmonary manifestations

## Abstract

Early recognition and prompt management are crucial for improving survival in COVID-19 patients, and after 2 years of the pandemic, many efforts have been made to obtain an early diagnosis. A key factor is the use of fast microbiological techniques, considering also that COVID-19 patients may show no peculiar signs and symptoms that may differentiate COVID-19 from other infective or non-infective diseases. These techniques were developed to promptly identify SARS-CoV-2 infection and to prevent viral spread and transmission. However, recent data about clinical, radiological and laboratory features of COVID-19 at time of hospitalization could help physicians in early suspicion of SARS-CoV-2 infection and distinguishing it from other etiologies. The knowledge of clinical features and microbiological techniques will be crucial in the next years when the endemic circulation of SARS-CoV-2 will be probably associated with clusters of infection. In this review we provide a state of the art about new advances in microbiological and clinical findings of SARS-CoV-2 infection in hospitalized patients with a focus on pulmonary and extrapulmonary characteristics, including the role of gut microbiota.

## 1. Introduction

Because of the pandemic caused by severe acute respiratory coronavirus-2 (SARS-CoV-2), the clinical understanding of the coronavirus disease 2019 (COVID-19) in humans remains to be further deepened [1,2]. As a matter of fact, early recognition and prompt management of hospitalized patients are crucial for improving survival, but COVID-19 infection may have not peculiar signs and symptoms that can differentiate this infection from other infective or non-infective causes of acute respiratory failure and/or fever [3].

In the last few months, a huge number of experiences have been published but despite this, much information about COVID-19 is still missing. Currently, all patients admitted to the emergency department with acute respiratory failure and/or fever should be considered as a suspected SARS-CoV-2 infection [4,5]. Of importance, the new multiplex molecular assays are able to screen samples from the respiratory tract for multiple pathogens simultaneously, using the syndromic approach. Moreover, in the last two years of the SARS-CoV-2 pandemic, the use of COVID-19 vaccines has deeply modified the natural evolution of this disease and its progression to severe pneumonia.

The main objective of this review is to describe new advances in clinical and laboratory findings of hospitalized patients with suspected COVID-19. An early diagnosis should be the milestone for prompt management of hospitalized COVID-19 patients to identify those that are candidates for an early administration of antiviral agents and/or monoclonal antibodies (MoAbs) in the first stages of the disease.

Based on the above, an early microbiological and clinical diagnosis of hospitalized patients with suspected COVID-19 is crucial for improving survival; moreover, the knowledge of pulmonary and extrapulmonary characteristics of patients is mandatory for managing and treating COVID-19 patients. Recent data showed that SARS-CoV-2 has a direct effect on gut microbiota. This interaction could induce a pro-inflammatory state and an alteration of amino acid transport with excessive gut permeability and microbial translocation.

## 2. Methods

The literature research was conducted using the well-established PubMed database. Case series, controlled and uncontrolled studies and meta-analyses were included. Case reports were also included. Only English-language papers were considered. We did not set a specific time window for the research, but the focus was placed on papers published in the last 2 years. The main keywords used were “SARS-CoV-2”, “COVID-19”, “microbiological diagnosis”, “clinical diagnosis”, “RT-PCR”, “fever”, “acute respiratory failure”, “mortality”, “gut microbiota” and “suspected COVID-19”.

In the first step, papers were screened by abstract and title. Then, the full text of the selected papers was examined. Papers were excluded if their content had nothing to do with COVID-19 pathophysiology, microbiological and/or clinical diagnosis. However, an important limitation of this review is that were no applied criteria for a systematic review of the literature.

## 3. New Advances in Microbiological Diagnosis of SARS-CoV-2 Infection

Since January 2020, several techniques have been developed to perform the correct diagnosis of SARS-CoV-2 infection [6]. The routine methods include molecular assay by real-time polymerase chain reaction (RT-PCR), to detect SARS-CoV-2 RNA in biological samples, and serological tests, to evaluate the concentration of IgM, IgG and IgA antibodies.

Even if several methods are being developed to detect SARS-CoV-2 in biological samples, such as reverse transcription loop-mediated isothermal amplification (RT-LAMP), clustered regularly interspaced short palindromic repeats (CRISPR) and droplet digital reverse transcription polymerase chain reaction (ddRT-PCR) [7], the RT-PCR is currently the gold standard for detecting the new virus from nasopharyngeal, oropharyngeal and nasal swabs and saliva. Among PCR-based methods, syndromic tests are frequently used for patients who exhibit more than one respiratory symptom to improve diagnosis and clinical management and to monitor possible respiratory co-infections [8]. However, molecular assays are not able to perform mass population screening. On the contrary, serological tests can be used to monitor antibody responses in individuals who received the SARS-CoV-2 vaccine and those who had a natural infection in a large cohort of the population [9].

Currently, to prevent viral spread and transmission, antigen tests appear as a very valuable tool to reach early and rapid detection of SARS-CoV-2, considering that therapies using new antiviral treatment could be started promptly within five days from symptom onset [10]. A limitation of these rapid tests is the possibility of obtaining a negative result in patients with low viral loads and infected by new SARS-CoV-2 variants. Despite the wide availability of diagnostic tests, the sensitivity is related to the cycle threshold (Ct) value regardless of vaccine status and/or symptoms. All negative results must be verified using molecular assay [11].

Serological and molecular assays may differ in sensitivity and specificity. Sensitivity is defined as the ability to detect very low SARS-CoV-2 viral loads or antibody titers accurately measured [8,12]. Specificity stands for the capacity to identify the specific SARS-CoV-2 genomic regions or IgA, IgM and IgG against the current pandemic pathogen without cross-reactions.

A summary of multiple approaches and their performances was reported in Table 1. “In-house” and non-certified diagnostic tests were excluded.

Very close to diagnostic methods, sequencing of the SARS-CoV-2 genome helps in understanding the pathogenicity and transmissibility of variants of concern (VOCs) and variants of interest (VOI) [13]. Indeed, alpha (B.1.1.7), beta (B.1.351), delta (B.1.617.2), gamma (P.1) and omicron (B.1.1.529) VOCs were related to an increase in transmissibility, severe diseases and reduced vaccine efficacy, while lambda (C.37) and mu (B.1.621) VOIs showed a reduced antibody neutralization and increased transmissibility of infection [14,15]. To date, omicron is becoming the more prevalent VOC worldwide; it is phylogenetically distinct from any known variants, and it is classified in several sublineages, BA.1, BA.2, BA.3, BA.4 and BA.5 [16].

The epidemiology of SARS-CoV-2 variants is continuously evolving, next-generation sequencing (NGS) and Sanger methods were applied to monitor the spread and viral evolution [17]. Sequencing of the whole genome or specific genomic regions, such as the S gene, is the only way to identify known and unknown variants highlighting a pattern of mutations [18]. Even if the NGS approach is expensive and not available in all diagnostic laboratories, it may detect mutations in several genomic regions affecting rapid antigen test results. The undetected variants can carry out deletion or substitutions in N-protein [19]. Additionally, rapid genotyping is useful for improving the prophylaxis with MoAbs of patients infected by SARS-CoV-2 viral variants [16]. Currently, the Stanford Coronavirus Resistance Database, an interactive tool that shows the neutralizing susceptibility information comparing SARS-CoV-2 sequences, is available for this purpose [20].

## 4. Clinical Diagnosis of COVID-19

Data from the literature highlighted peculiar characteristics associated with SARS-CoV-2 etiology in suspected COVID-19 patients in the emergency department: a prompt identification of specific clinical characteristics (such as acute dyspnea, dry cough and/or duration of fever >3 days), and laboratory findings (such as lymphocytopenia and serum ferritin) can help physicians to distinguish between COVID-19 and other etiologies [21].

As previously reported, the application of a standard approach to the management of patients with acute respiratory failure and/or fever [5,22] associated with the knowledge of clinical and laboratory characteristics of COVID-19 can drive physicians to early therapeutic choices. Of importance, data confirmed that age ≥ 65 years and intensive care unit (ICU) admission are independently associated with all-cause 30-day mortality [3,4], while the use of steroids, low-molecular-weight heparin (LMWH), MoAbs and antiviral agents (such as molnupiravir, remdesivir or nirmatrelvir/ritonavir) were associated with lower rates of hospitalization and mortality [23,24].

Of interest, data from a large cohort of patients in Moscow [25] showed that typical fever, fatigue, cough and shortness of breath were the most frequent symptoms observed in suspected COVID-19 patients, in agreement with other studies in other countries [26,27]. However, it had been highlighted that every variant of the virus was selected during the different epidemic waves. These variants were characterized by changes in several viral proteins, reducing the efficacy of treatments such as MoAbs, and also a potential increase in disease severity and/or transmissibility or different clinical manifestations [28]. The WHO differentiated the novel SARS-CoV-2 strains in VOIs and VOCs. Of importance, VOCs were associated with a reduced susceptibility to the MoAbs and vaccination; interestingly, omicron VOC showed a higher transmissibility and a higher risk of re-infection, as reported below [29].

Regarding the clinical characteristics of COVID-19 patients, fever, fatigue, dry cough, dyspnea, myalgia, breathlessness and/or severe acute distress respiratory syndrome have been recognized as the most frequent in the first stages of the disease. However, the spreading of new VOCs, such as omicron, determined different clinical conditions involving mainly the upper respiratory tract [30]. This could be related to specific mutations that cause a modification in the viral interaction sites prevalently located in the upper airways [31]. In fact, some studies show how the omicron variant mainly replicates more rapidly in the bronchi than in the lung parenchyma [32] and how it replicates less in the lower airway tract compared to the delta variant [33]. Some studies on populations affected by the omicron variant highlighted how symptoms were characterized by a more moderate clinical presentation than the population affected by other variants of SARS-CoV-2 (dry cough, headache, asthenia and dyspnea), and lower respiratory airways are involved less frequently. These data were also confirmed by the lower rate of hospitalization, recovery in an intensive setting, the need for invasive and non-invasive mechanical ventilation and mortality during hospitalization compared to patients affected by different variants of SARS-CoV-2 [34,35].

It is important to underline, however, how these data may be affected by the greater vaccination coverage of the population compared to previous waves in which the vaccination campaign was still ongoing. In fact, in the group of patients with a complete vaccination course, a lower rate of need for non-invasive ventilation was observed, and stratifying by age, there were no statistically significant differences [34]. This, therefore, confirms the need to continue the vaccination campaign, despite a less severe clinical presentation related to omicron VOC.

Another important aspect to consider is re-infections; it was shown that the emergence of the omicron variant, and more recently the omicron-2 variant, is associated with several cases of re-infection that occurred even in vaccinated subjects. Re-infection is defined as follows: (1) persons with at least one detection of SARS-CoV-2 test, more than 90 days after the first detection of SARS-CoV-2, whether or not symptoms were present, and (2) persons with COVID-19-like symptoms and detection of SARS-CoV-2 RNA. Currently, poor data are available and highlighted as a majority of re-infected patients are asymptomatic or paucisymptomatic (resumption of respiratory or gastrointestinal symptoms) [36,37]. Hospitalization was required only in some cases at the time of re-infection, but this condition was mainly necessary for complications related to multiple comorbidities, such as diabetes mellitus, immunodeficiency or iatrogenic immunosuppression, as well as the age of patients rather than COVID-19-related symptoms [38,39]. In a subgroup analysis, it was found that the prevalence of re-infection is higher in male than in female subjects.

Of interest, recurrence or reactivation is defined as the manifestation of symptoms after hospitalization and refers to an infection of the same species and strain of SARS-CoV-2 that can occur due to host immunodeficiency [40]. As reported for re-infections, most patients with reactivation are asymptomatic or paucisymptomatic with comorbidities including diabetes mellitus, hypertension, chronic pulmonary disease and cardiovascular disease. Of importance, these underlying diseases may confuse the differential diagnosis of COVID-19 patients from other diseases, based only on clinical symptoms. Therefore, it is important to better understand information about the prevalence, risk factors, comorbidities and anamnestic data of the patient as they allow orienting the clinician toward the risk of re-infection or reactivation or the related risk for hospitalization.

An important aspect of SARS-CoV-2 is related to laboratory alterations. The most common laboratory test alterations are as follows: lymphocytopenia, increase in transaminases, C-reactive protein, interleukin-6 (IL-6), CPK, LDH, prothrombin time, ferritin, creatinine and D-dimer. Of importance, in the pediatric population, a reduction in lymphocyte counts is reported in about 16% of cases [41], while alteration of CK-MB values is reported in 37% of cases; out of these, over 80% of these alterations were observed in children <1 year of age. This evidence may suggest greater attention to myocardial damage in the pediatric population, especially in children <1 year of age. Of importance, children may develop severe inflammatory syndrome, with symptoms similar to Kawasaki disease or toxic shock syndrome. This condition is called multisystem inflammatory syndrome in children (MIS-C) [42].

Lymphocytopenia has been previously reported to be associated with higher severity and mortality in COVID-19 patients [43,44]. Of interest, data from an Italian cohort demonstrated that ferritin levels over the 25th percentile are associated with a more severe pulmonary involvement, independently of age and gender [45].

As a matter of fact, CT findings in COVID-19 patients were frequently indistinguishable from other etiologies, such as acute heart failure, bacterial pneumonia or pulmonary embolism [5,21]. Considering typical radiological findings in COVID-19, it is important to underline that this population, especially in the first stage, could present only extrapulmonary manifestations with a normal CT finding [46]. In this scenario, it is important to underline that a negative RT-PCR test result does not exclude the possibility of COVID-19, and repeated testing and sampling were shown to improve the sensitivity of RT-PCR [47]. Thus, a rigorous application of a standard approach to the management of suspected COVID-19, associated with clinical and laboratory findings, could help physicians in the emergency department.

Finally, a peculiar aspect of COVID-19 patients from the “second wave” of the pandemic was the widespread use of steroids, LMWH and remdesivir that could have modified the outcomes of this population [48]. A meta-analysis including seven RCTs [49,50,51,52,53,54] showed that the use of corticosteroids compared with usual care or placebo lowered the 28-day all-cause mortality [55], especially in patients receiving invasive mechanical ventilation [56,57]. Conversely, in patients with mild COVID-19, steroid treatment was associated with a worse clinical outcome [58], a higher risk of progression to severe disease and a prolonged hospital stay [59], highlighting the main role of steroid treatment prevalently in moderate to severe COVID-19. Important observations were reported about the role of LMWH in the survival of COVID-19 patients, especially in reducing thromboembolism complications in critically ill patients [60]. Of interest, different data were reported about worldwide the efficacy of remdesivir, taking into account different outcomes [61,62]. These data showed an important role for remdesivir in the early stage of the disease, to reduce progression to severe pneumonia.

A proposed system of management for patients with suspected COVID-19 at hospital admission is reported in Figure 1.

## 5. Extrapulmonary Manifestations and the Role of Gut Microbiota in COVID-19

Extrapulmonary manifestations can often occur prior to respiratory symptoms and could represent an important indicator for early clinical suspicion of COVID-19 [63]. Among these are included a broad spectrum of disorders with gastrointestinal, neurological, coagulation, renal, endothelial, cardiovascular and hepatobiliary involvement that are associated with increased mortality risk and prolonged hospitalization. Extrapulmonary damage may be associated with direct viral damage, cytokine cascade induced by systemic inflammatory response or endothelial damage [64].

Cardiovascular complications can occur in patients with heart failure or coronary artery disease. One of the most common is myocarditis that may be characterized by myocardial damage in the absence of ischemic causes and inflammatory infiltrates [65,66]. In almost a third of COVID-19 patients with myocardial involvement, cardiogenic shock occurred [67] with a mortality of about 26% [68]. In addition, COVID-19 patients have a high risk of arterial and venous thrombosis [69].

Other manifestations can involve other districts. Kidney damage is a frequent complication in hospitalized patients with COVID-19 with an incidence range between 10 and 80% and a significantly increased risk of intra-hospital mortality [70,71]. Neurological symptoms occur most commonly in patients with severe forms of COVID-19 [72,73]. COVID-19 is reported to be associated with an increased risk of mental disorders such as depression, anxiety, schizophrenia, phobia, obsessive–compulsive disorder and post-traumatic stress disorder. Loss of smell and taste was observed in 40% of COVID-19 patients [74]. A European multicenter study reported that about 12% of patients had ageusia and anosmia as early symptoms [75]. Ocular manifestations are often non-specific, and conjunctivitis, mainly manifested as redness, watering, discharge and foreign body sensations, is the most frequently reported; other ocular complications include dry eye, blurred vision, ocular pain, photophobia and itchiness [76]. Notably, ocular signs and symptoms were the initial presentations in 3.3% of COVID-19 patients [76]. Dermatological manifestations have been observed in 2–20% of patients [77,78]. The most common manifestations are maculopapular rash, urticaria, petechiae/purpura, blisters, chilblains, livedo racemose, and necrosis or distal ischemia [79]. Most of these skin lesions are self-limiting and do not appear to be associated with the severity of the disease.

Several COVID-19 patients present gastrointestinal symptoms, principally vomiting and diarrhea, and the severity rate was more than 40% in these patients [80]. The presence of gastrointestinal symptoms is associated with a more severe disease. It has been shown that the presence of nausea, vomiting, diarrhea and abdominal pain can precede or accompany the classic respiratory symptoms with an incidence between 10 and 60% [81]. This is relevant, as 10% of patients initially present only gastrointestinal symptoms without any respiratory symptoms, resulting in a possible delay in the diagnosis of COVID-19 [82]. In a meta-analysis of Cheung et al. that included 60 studies and 4243 patients, the three most common symptoms were anorexia, diarrhea, and nausea or vomiting [83]. The presence of such symptoms was associated with a high risk of ARDS, the need for non-invasive ventilation and orotracheal intubation, but not mortality [84]. Of importance, the prevalence of liver damage is around 15–55% [85,86], with an incidence rate of liver damage in deceased patients of 78%. Alteration of AST values shows a strong correlation with mortality compared to other liver damage indices such as ALT, TBIL and ALP [87,88]. Kaafarani et al. observed that of 141 critically ill patients, during hospitalization in intensive care, 4% developed acalculous cholecystitis and 1% developed pancreatitis [83,89].

Data show an important role of gut microbiota in COVID-19 patients [87]. Correlation between the immune system and gut microbiota has been demonstrated in germ-free mice, which presented a deficiency in regulatory T cells [90]. Recent studies make it evident that gut microbiota can modulate neutrophil migration and T-cell differentiation in subpopulations [91]. Viral infections change gut microbiota: reduction in lactobacilli and firmicutes leads to a dysbiosis that drives a dysfunction of the normal role of microbiota [92]. Angiotensin-converting enzyme 2 receptor is well represented in the intestinal tract [93] and acts as a functional receptor for SARS-CoV-2 [94]. This interaction could induce a pro-inflammatory state and an alteration of amino acid transport with excessive gut permeability and microbial translocation [80,95].

Of importance, probiotic supplementation is also recognized as an important therapeutic strategy in COVID-19 patients. Probiotic supplementation can maintain the enterocytic barrier [96]. In an open-label parallel-group trial, in a single center, by d’Ettorre et al., a probiotic multistrain modulation (streptococcus and lactobacillus) administration (2.4 × 10⁶ UFC/day for 14 days) in adjunction to standard therapy reduced gastrointestinal symptoms, mortality and intensive care unit transfer in hospitalized COVID-19 patients [97]. It is important to underline that a correct state of gut microbiota promotes reduced inflammation, and thus a better prognosis for the patient [98]. Thus, probiotic supplementation could have a beneficial role in COVID-19 patients with gastrointestinal symptoms. The limited expenditure for the treatment is a further factor to consider [97]. The most common strains commercially available belong to the *Lactobacillus* and *Bifidobacterium* species, which proved some beneficial effects on the human body when administered in adequate amounts. These mentioned bacterial species are known to be involved in some essential physiological functions such as stimulation of immune response, prevention of pathogenic and opportunistic microbial colonization, production of SCFA, catabolism of carcinogenic substances and synthesis of vitamins such as B and K. Thus, as part of post-acute COVID-19 syndrome, the dysbiosis of the gut microbiota could potentially become a target for evaluation of the state of health and consequentially for targeted intervention.

Immune-modulatory effects of probiotics might be relevant to preventing complications of COVID-19. The dysbiosis in COVID-19 patients may play a relevant role in determining the course of the disease by increasing systemic pro-inflammatory cytokine production. Microbiota could have therapeutic properties by reducing gastrointestinal symptoms. Although the underlying mechanism for the observed dysbiosis is unclear, it might happen via downregulation of ACE2 expression that alleviates the intestinal absorption of tryptophan leading to decreased secretion of antimicrobial peptides and changing the composition of gut microbiota. As a promising strategy, it might not only improve the quality of life of patients but also reduce the burden on the public health system because it will prevent the onset of comorbidities linked to dysbiosis of the gut microbiota [99,100].

The literature about COVID-19 and the microbiota is in a continuous evolution [99,100,101,102]. Many significant studies are published daily, especially on the aspects discussed in this review. Future studies are needed to improve our knowledge about the most important determinants of this disease. Of importance, the gut and respiratory tract microbiome may change in COVID-19 patients with opportunistic pathogen abundance. Further research should be conducted to elucidate the effect of alternation of the human microbiome in disease pathogenesis and progression to severe stages [103,104].

## 6. Conclusions

In conclusion, COVID-19 syndrome is characterized by a heterogeneous clinical, laboratory and radiological presentation, especially at its onset. However, data showed that patients with a confirmed diagnosis of COVID-19 can show peculiar characteristics at the time of hospitalization that could help physicians to distinguish COVID-19 from other infective or non-infective etiologies [105,106,107].

Considering the application of a standard approach to the management of patients with acute respiratory failure and/or fever, the knowledge of clinical and laboratory characteristics of COVID-19 can drive early therapeutic choices [108,109,110,111,112,113,114,115,116,117,118,119,120,121,122]. Of interest, in the management of extrapulmonary manifestations, mainly represented by gastrointestinal symptoms, the use of probiotic supplementation could be evaluated.

## Figures and Tables

**Figure 1 diagnostics-12-02143-f001:**
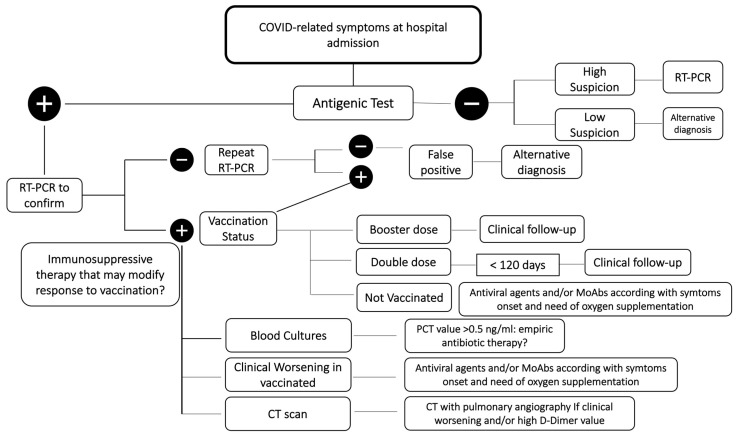
Management of patients with suspected COVID-19. Legend: RT-PCR: real-time polymerase chain reaction; CT: computed tomography; MoAbs: monoclonal antibodies; PCT: procalcitonin.

**Table 1 diagnostics-12-02143-t001:** Certified diagnostic methods routine used in clinical practice.

Test Name	Technology	Manufacturer	Specimen	Sensitivity% (95%; C.I.)	Specificity% (95%; C.I.)	TAT (min)	Reference
**Molecular assay**							
ARGENE SARSCOV-2 RGENE (*RdRp, N, E*)	Two triplex PCR	bioMe’rieux (France)	Nasopharyngeal swabs	NA	NA	3–4 h	Lai C-C et al., 2021 [8]
Allplex 2019-nCoV assay Target genes (*E, N* and *RdRp*)	Multiplex real-time PCR	Seegene (Korea)	Sputum, nasopharyngeal swab, nasopharyngeal aspirate, bronchoalveolar lavage, throat swab	98.2 (95%; 90.3–100.0)	100 (95%; 94.9–100.0)	110	Zhou Y et al., 2021 [3]
TaqPath COVID-19 high throughput combo Target genes (*S, N*)	Multiplex real-time PCR	Thermo Fisher Scientific (USA)	Saliva, nasal, nasopharyngeal samples	100 (95%; 97.9–100.0)	100 (95%; 98.6–100)	Can run up to 8000 reactions per day	Zhou Y et al., 2021 [3]
Xpert Xpress SARS-CoV-2 Target genes (*N2, E*)	RT-PCR (point of care)	Cepheid (USA)	Nasopharyngeal swabs, nasal aspirate	99 (95%; 97.0–99.0)	97 (95%; 95.0–98.0)	45	Lai C-C et al., 2021 [8]
ID NOW COVID-19 assay Target gene (*RdpRp*)	Isothermal nucleic acid amplification	Abbot (USA)	Nasopharyngeal and nasal swabs	95.0%	97.9%	≤13	Lai C-C et al., 2021 [8]
iAMP COVID-19 detection kit Target genes (*N, ORF1ab*)	Isothermal real-time fluorescent reverse transcription	Atilia BioSystems (USA)	Nasopharyngeal, oropharyngeal swabs	NA	NA	90	Guaman-Bautista et al., 2021 [7]
Agilent SARS-CoV-2 qRT-PCR Dx Kit Target genes (N1 and N2)	Real-time reverse transcription-polymerase chain reaction	Agilent	Nasopharyngeal, nasal and oropharyngeal swabs	95.0%	NA	90	CDC
**Syndromic test**							
BiofireFilmarrayRP.-2.1 Target genes (*RdRp, N, E*)	Multiplex real-time PCR	bioMérieux (France)	Nasopharyngeal swabs	98.4 (NA)	98.9 (NA)	45	Lai C-C et al., 2021 [8]; https://www.fda.gov
QIAstat-Dx Respiratory 2019-nCoV Panel (22 targets) (*RdRp, E*)	Multiplex real-time PCR	QIAGEN (Netherlands)	Nasopharyngeal swabs	97.2 (NA)	96.1 (NA)	<70 min	Lai C-C et al., 2021 [8]; https://www.qiagen.com
**Serological assay**							
SARS-CoV-2 IgG (CMIA) IgG	Chemiluminescent microparticle immunoassay	Abbott Laboratories, IL, USA	Serum or plasma	3–7 days after symptom onset: 25.0 8–13 days: 86.1 >14 days: 100	99.6–100	NA	Lai C-C et al., 2021 [8]
Elecsys Anti-SARSCoV-2 (ECLIA) Total antibody (including IgG)	Electrochemiluminescence	Roche Diagnostics Basel, Switzerland	Serum or plasma	0–6 days aftersymptom onset: 65.5 7–13 days: 88.1 >14 days: 100	99.8	18 min	Lai C-C et al., 2021 [8]
Anti-SARS-CoV-2 ELISA (EIA) IgA and IgG	Immunoenzymatic	EUROIMMUN AG (Germany)	Serum	NA	NA	2–3 h/96 samples	Lai C-C et al., 2021 [8]
LIAISON SARS-CoV-2 S1/S2 IgG (CLIA) IgG	Chemiluminescence	DiaSorin	Serum	97.56	99.3	35	Zhou Y et al., 2021 [8]
Platelia SARS-CoV-2 total antibody assay (ELISA) Total antibody	Immunoenzymatic	Bio-Rad	Serum	100.0	99.6	NA	Zhou Y et al., 2021 [8]
Access SARS-CoV-2 IgG(CLIA)	Chemiluminescence	Beckman Coulter, Inc.	Serum	96.8	99.6	NA	Guaman-Bautista LP., et al., 2021 [7]

## Data Availability

Not applicable.

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
