# Peer review of "Microbiological and Clinical Findings of SARS-CoV-2 Infection after 2 Years of Pandemic: From Lung to Gut Microbiota"

_diagnostics, 2022, doi:10.3390/diagnostics12092143_

Round 1
Reviewer 1 Report (New Reviewer)
The article summarizes the clinical and microbiological knowledge well and was compiled two years after the first cases of SARS-COV 2 infection. They rightly emphasize that
My concerns:
- Lack of methodology for the selection of scientific literature used to prepare the manuscript,
- Too sketchy chapter on gut microbiota. What changes in the microbiome prevail in post-infection patients, do they persist for several weeks or months? There is only a following sentence: „reduction of lactobacilli and firmicutes leading to a dysbiosis that drive to a disfunction of the normal role of microbiota?”
- Is there an individual microbiological signature in COVID-19 patients?
- The Authors did not submit the „Limitations of the study”
Author Response
The article summarizes the clinical and microbiological knowledge well and was compiled two years after the first cases of SARS-COV 2 infection. They rightly emphasize that
My concerns:
- Lack of methodology for the selection of scientific literature used to prepare the manuscript,
R: dear reviewer, thank you very much for these comments. About methodology, this is a narrative review, then were not applied criteria for systematic review of the literature. However, we included a methods section.
- Too sketchy chapter on gut microbiota. What changes in the microbiome prevail in post-infection patients, do they persist for several weeks or months? There is only a following sentence: „reduction of lactobacilli and firmicutes leading to a dysbiosis that drive to a disfunction of the normal role of microbiota?”
R: we modified section about microbiota to deeply discuss its role in COVID-19 patients.
- Is there an individual microbiological signature in COVID-19 patients?
R: No definitive data are available abotu an individual microbiological signature in COVID-19 patients. To date, the individual signature of COVID-19 patient is a combination of microbiological (presence of natural antibodies or antibodies after vaccination) and clinical manifestations (asyntomatic, milde or severe symptoms). We reported a brief sentence in section 5.
- The Authors did not submit the „Limitations of the study”
R: thanks also for this comment. We included a limitation paragraph in the methods section.
Reviewer 2 Report (New Reviewer)
This article describes after the COVID-19 pandemic,in addition to traditional microbiological detection techniques,how data on clinical, imaging, and laboratory characteristics of COVID-19 at the time of hospitalization can help to early suspect SARS-CoV-2 infection and distinguish it from other etiologies. This article reviews the microbiological and clinical research progress of hospitalized patients infected with SARS-CoV-2, focusing on pulmonary and extrapulmonary characteristics, including the role of intestinal microbiota, which are innovative and have certain effects on improving the survival of patients with COVID-19. However, there are still some problems which need to be solved before it is considered for publication.
1. There is a paradox between line 15 and line 33, whether the COVID-19 have peculiar signs and symptoms is less clear in the article.
2. The efficacy of steroids, LMWH, and remdesivir is described in line 181 to 191. It is relatively detailed in the part of Steroids but not in LMWH and remdesivir one. Additional clarification is recommended.
3. The current gold standard for the diagnosis of SARS-CoV-2 is RT-PCR. The author mentions that A limitation of these rapid tests is the possibility to obtain a negative result in patients with low viral loads and infected by new SARS-CoV-2 variants. But its timeliness may be stronger than the clinical observation, so can the interpretation of clinical symptoms be used as an auxiliary diagnostic method? Or is the gold standard better than clinical symptom observation in the goal of early diagnosis? The connection between the methods is recommended to supplement.
4. Considering the influence of individual differences on the clinical symptoms of SARS-CoV-2 infection, this article only summarize the symptoms and indicators that are currently more obvious in COVID-19 patients to distinguish them from other diseases, but there are still doubts about their help for diagnosis specificity.
5. As reported for re-infections, most patients with reactivation are asymptomatic or pauci-symptomatic with comorbidities including diabetes mellitus, hypertension, chronic pulmonary disease, cardiovascular disease. Will the underlying disease of the patient confuse the differential diagnosis of COVID-19 and other diseases based on clinical symptoms?
Author Response
Reviewer 2
This article describes after the COVID-19 pandemic,in addition to traditional microbiological detection techniques,how data on clinical, imaging, and laboratory characteristics of COVID-19 at the time of hospitalization can help to early suspect SARS-CoV-2 infection and distinguish it from other etiologies. This article reviews the microbiological and clinical research progress of hospitalized patients infected with SARS-CoV-2, focusing on pulmonary and extrapulmonary characteristics, including the role of intestinal microbiota, which are innovative and have certain effects on improving the survival of patients with COVID-19. However, there are still some problems which need to be solved before it is considered for publication.
R: dear reviewer, we are grateful for your comments to improve quality of our manuscript. We addressed all your suggestions.
- There is a paradox between line 15 and line 33, whether the COVID-19 have peculiar signs and symptoms is less clear in the article.
R: we corrected it in the manuscript.
- The efficacy of steroids, LMWH, and remdesivir is described in line 181 to 191. It is relatively detailed in the part of Steroids but not in LMWH and remdesivir one. Additional clarification is recommended.
R: we modified the sentence to comment also LMWH and remdesivir, considering that a deep discussion of therapeutic management is out of the scope of this review.
- The current gold standard for the diagnosis of SARS-CoV-2 is RT-PCR. The author mentions that A limitation of these rapid tests is the possibility to obtain a negative result in patients with low viral loads and infected by new SARS-CoV-2 variants. But its timeliness may be stronger than the clinical observation, so can the interpretation of clinical symptoms be used as an auxiliary diagnostic method? Or is the gold standard better than clinical symptom observation in the goal of early diagnosis? The connection between the methods is recommended to supplement.
R: To confirm clinical diagnosis is important to obtain a SARS-CoV-2 positive results by molecular assay, defined as gold standard, from one respiratory sample (such as nasopharyngeal swabs, saliva, and so on) of patient. To supplement the connection between the diagnostic methods and clinical symptoms, we added few sentences. For an early diagnosis to exclude other respiratory pathogens, responsible of same symptoms, we introduced in Introduction section and discussed in Section 2 the syndromic tests. Additionally, we implemented Table 1, including specimen used, and we added the importance of serological test.
- Considering the influence of individual differences on the clinical symptoms of SARS-CoV-2 infection, this article only summarize the symptoms and indicators that are currently more obvious in COVID-19 patients to distinguish them from other diseases, but there are still doubts about their help for diagnosis specificity.
R: we totally agree with this comment. However, the scope of this review was to discuss, after 2 years, symptoms and indicators (mainly represented by laboratory findings) that could help physicians (for example in an emergency department) to detect patients with a higher risk of COVID-19 and unfavorable progression.
- As reported for re-infections, most patients with reactivation are asymptomatic or pauci-symptomatic with comorbidities including diabetes mellitus, hypertension, chronic pulmonary disease, cardiovascular disease. Will the underlying disease of the patient confuse the differential diagnosis of COVID-19 and other diseases based on clinical symptoms?
R: another important comment. We included it in that paragraph.
Reviewer 3 Report (New Reviewer)
Microbiological and clinical findings of SARS-CoV-2 infection 2 after 2 years of pandemic: from lung
to gut microbiota
General comments
1. Section 1 of the introduction needs a general introduction about gut microbiota, how
it comes into play
2. PCR has not introduced well as it has focused only on RT-PCR while there are other
types or variations of PCR such as qRT-PCR, d-PCR, point-of care PCR…..
3. This a review paper that claims to “provide a state-of-the-art about new advances in
microbiological and clinical findings of SARS-CoV-2 infection”. However, the paper
appears to be missing a lot of important and most recent references. The literature
search was not thorough enough. Here are some papers that are missing, just to
mention a few:
3.1 Pham, V.H.; Gargiulo Isacco, C.; Nguyen, K.C.D.; Le, S.H.; Tran, D.K.; Nguyen,
Q.V.; Pham, H.T.; Aityan, S.; Pham, S.T.; Cantore, S.; et al. Rapid and sensitive
diagnostic procedure for multiple detection of pandemic coronaviridae family
members SARS-CoV-2, SARS-CoV, MERS-CoV and HCoV: A translational
research and cooperation between the Phan Chau Trinh University in Vietnam
and university of Bari “Aldo Moro” in Italy. Eur. Rev. Med. Pharmacol. Sci. 2020,
24, 7173–719
3.2 Ai, T., Yang, Z., Hou, H., Zhan, C., Chen, C., Lv, W., Tao, Q., Sun, Z. and Xia, L.,
2020. Correlation of chest CT and RT-PCR testing in coronavirus disease 2019
(COVID-19) in China: a report of 1014 cases. Radiology.
3.3 Liu, Q., Mak, J.W.Y., Su, Q., Yeoh, Y.K., Lui, G.C.Y., Ng, S.S.S., Zhang, F., Li, A.Y.,
Lu, W., Hui, D.S.C. and Chan, P.K., 2022. Gut microbiota dynamics in a
prospective cohort of patients with post-acute COVID-19
syndrome. Gut, 71(3), pp.544-552.
3.4 Mizutani, T., Ishizaka, A., Koga, M., Ikeuchi, K., Saito, M., Adachi, E., Yamayoshi,
S., Iwatsuki-Horimoto, K., Yasuhara, A., Kiyono, H. and Matano, T., 2022.
Correlation Analysis between Gut Microbiota Alterations and the Cytokine
Response in Patients with Coronavirus Disease during Hospitalization.
Microbiology Spectrum, 10(2), pp.e01689-21.
3.5 Vestad, B., Ueland, T., Lerum, T.V., Dahl, T.B., Holm, K., Barratt‐Due, A., Kåsine,
T., Dyrhol‐Riise, A.M., Stiksrud, B., Tonby, K. and Hoel, H., 2022. Respiratory
dysfunction three months after severe COVID‐19 is associated with gut
microbiota alterations. Journal of internal medicine, 291(6), pp.801-812.
3.6 Brumfield, K.D., Leddy, M., Usmani, M., Cotruvo, J.A., Tien, C.T., Dorsey, S.,
Graubics, K., Fanelli, B., Zhou, I., Registe, N. and Dadlani, M., 2022. Microbiome
Analysis for Wastewater Surveillance during COVID-19. mBio, pp.e00591-22.
4. Section 2 needs to be made more comprehensive, backed with the temporal
biochemical aspects of the infection and the pathophysiology of the disease showing
when and where samples are taken and how diagnosis is conducted. Admittedly,
many aspects of this have been discussed in Section 3, indicating that Section 3 is more
thorough than Section 2. In particular, Section 2 is missing some methods that have
been discussed in other papers with a similar title to the title of the section. Eg Guo,
J., Ge, J. and Guo, Y., 2022. Recent advances in methods for the diagnosis of Corona
Virus Disease 2019. Journal of Clinical Laboratory Analysis, 36(1), p.e24178.
Specific comments
1. Line 55 indicates that RT-PCR detects the presence of viruses from nasopharyngeal,
oropharyngeal, nasal swabs and saliva. The authors expounded on PCR but presented
little on serological tests. What specimens are used in serological tests?
2. Line 64: Please define what the terms sensitivity and specificity stand for
3. Table 1 has a section called Syndromic tests and yet this term has not been introduced
in the introduction, nor has it been discussed in the Discussion section.
4. In Table 1, it will help if the specimen used is added, eg blood, saliva, nasal swabs etc
5. Is it possible to link the effect of Covid 19 on gut microbiota to waste water
epidemiology of the disease? In fact there are a few papers on this, such as Paterson,
B.J. and Durrheim, D.N., 2022. Wastewater surveillance: an effective and adaptable
surveillance tool in settings with a low prevalence of COVID-19. The Lancet Planetary
Health, 6(2), pp.e87-e88. In fact Barua et al 2022 (Tracking the temporal variation of
COVID-19 surges through wastewater-based epidemiology during the peak of the
pandemic: A six-month long study in Charlotte, North Carolina. Science of the Total
Environment, 814, p.152503) reported that SARS-CoV-2 in wastewater had 5–12 days
lead time than clinical COVID-19 cases. Seeing the paper is on “microbiological and
clinical findings of SARS-CoV-2 infection, perhaps that needs to be discussed too.
6. What do the names in lines 459-563 represent?
Author Response
Microbiological and clinical findings of SARS-CoV-2 infection 2 after 2 years of pandemic: from lung to gut microbiota
R: dear reviewer, thanks very much for all your important comments. We think that now quality of our manuscript is improved.
General comments
- Section 1 of the introduction needs a general introduction about gut microbiota, how it comes into play
R: we modified as required.
- PCR has not introduced well as it has focused only on RT-PCR while there are other types or variations of PCR such as qRT-PCR, d-PCR, point-of care PCR…..
R: To give complete information about molecular assays used in the diagnostic procedures, we included the "technology” column in Table 1. We added methods reported in “Eg Guo, J., Ge, J. and Guo, Y., 2022. Recent advances in methods for the diagnosis of Corona Virus Disease 2019. Journal of Clinical Laboratory Analysis, 36(1), p.e24178.”.
- This a review paper that claims to “provide a state-of-the-art about new advances in microbiological and clinical findings of SARS-CoV-2 infection”. However, the paper appears to be missing a lot of important and most recent references. The literature search was not thorough enough. Here are some papers that are missing, just to mention a few:
- Pham, V.H.; Gargiulo Isacco, C.; Nguyen, K.C.D.; Le, S.H.; Tran, D.K.; Nguyen, Q.V.; Pham, H.T.; Aityan, S.; Pham, S.T.; Cantore, S.; et al. Rapid and sensitive diagnostic procedure for multiple detection of pandemic coronaviridae family members SARS-CoV-2, SARS-CoV, MERS-CoV and HCoV: A translational research and cooperation between the Phan Chau Trinh University in Vietnam and university of Bari “Aldo Moro” in Italy. Eur. Rev. Med. Pharmacol. Sci. 2020, 24, 7173–719
- Ai, T., Yang, Z., Hou, H., Zhan, C., Chen, C., Lv, W., Tao, Q., Sun, Z. and Xia, L., 2020. Correlation of chest CT and RT-PCR testing in coronavirus disease 2019 (COVID-19) in China: a report of 1014 cases. Radiology.
- Liu, Q., Mak, J.W.Y., Su, Q., Yeoh, Y.K., Lui, G.C.Y., Ng, S.S.S., Zhang, F., Li, A.Y., Lu, W., Hui, D.S.C. and Chan, P.K., 2022. Gut microbiota dynamics in a prospective cohort of patients with post-acute COVID-19 syndrome. Gut, 71(3), pp.544-552.
- Mizutani, T., Ishizaka, A., Koga, M., Ikeuchi, K., Saito, M., Adachi, E., Yamayoshi, S., Iwatsuki-Horimoto, K., Yasuhara, A., Kiyono, H. and Matano, T., 2022. Correlation Analysis between Gut Microbiota Alterations and the Cytokine Response in Patients with Coronavirus Disease during Hospitalization. Microbiology Spectrum, 10(2), pp.e01689-21.
- Vestad, B., Ueland, T., Lerum, T.V., Dahl, T.B., Holm, K., Barratt‐Due, A., Kåsine, T., Dyrhol‐Riise, A.M., Stiksrud, B., Tonby, K. and Hoel, H., 2022. Respiratory dysfunction three months after severe COVID‐19 is associated with gut microbiota alterations. Journal of internal medicine, 291(6), pp.801-812.
- Brumfield, K.D., Leddy, M., Usmani, M., Cotruvo, J.A., Tien, C.T., Dorsey, S., Graubics, K., Fanelli, B., Zhou, I., Registe, N. and Dadlani, M., 2022. Microbiome Analysis for Wastewater Surveillance during COVID-19. mBio, pp.e00591-22.
R: dear reviewer, this is a narrative review. Then, we selected some study on the basis of our decision. However, you included interesting studies then we cited it in the text also to have an updated manuscript.
- Section 2 needs to be made more comprehensive, backed with the temporal biochemical aspects of the infection and the pathophysiology of the disease showing when and where samples are taken and how diagnosis is conducted. Admittedly, many aspects of this have been discussed in Section 3, indicating that Section 3 is more thorough than Section 2. In particular, Section 2 is missing some methods that have been discussed in other papers with a similar title to the title of the section. Eg Guo, J., Ge, J. and Guo, Y., 2022. Recent advances in methods for the diagnosis of Corona Virus Disease 2019. Journal of Clinical Laboratory Analysis, 36(1), p.e24178.
R: The purpose of the Section 2 is to list the microbiological methods that can be applied in the diagnosis of positive subjects. Timing and clinical applications are discussed in Section 3, dedicated to the clinical part. In Section 2, we added methods reported in “Eg Guo, J., Ge, J. and Guo, Y., 2022. Recent advances in methods for the diagnosis of Corona Virus Disease 2019. Journal of Clinical Laboratory Analysis, 36(1), p.e24178.”. Additionally, to give complete informations, we included the "technology” column in Table 1.
Specific comments
- Line 55 indicates that RT-PCR detects the presence of viruses from nasopharyngeal, oropharyngeal, nasal swabs and saliva. The authors expounded on PCR but presented little on serological tests. What specimens are used in serological tests?
R: We added the importance of serological tests using the following reference „Muecksch F, Wise H, Templeton K, Batchelor B, Squires M, McCance K, Jarvis L, Malloy K, Furrie E, Richardson C, MacGuire J, Godber I, Burns A, Mavin S, Zhang F, Schmidt F, Bieniasz PD, Jenks S, Hatziioannou T. Longitudinal variation in SARS-CoV-2 antibody levels and emergence of viral variants: a serological analysis. Lancet Microbe. 2022 Jul;3(7):e493-e502. doi: 10.1016/S2666-5247(22)00090-8”. Additionally, in Table 1 we added the “specimen” column to specify sample used in serological assays.
- Line 64: Please define what the terms sensitivity and specificity stand for
R: We added the definitions of sensitivity and specificity using the following reference “Scarcella S, Rizzelli A, Fontana A, Zecca C, Pasanisi G, Musio K, Putignano AL, Aprile V, Fedele A, Errico P, Copetti M, Tassi V. A CLEIA Antigen Assay in Diagnosis and Follow-Up of SARS-CoV-2-Positive Subjects. Microbiol Spectr. 2022 Jun 29;10(3):e0103221. doi: 10.1128/spectrum.01032-21.” and “Cassidy H, van Genne M, Lizarazo-Forero E, Niesters HGM, Gard L. Evaluation of the QIAstat-Dx RP2.0 and the BioFire FilmArray RP2.1 for the Rapid Detection of Respiratory Pathogens Including SARS-CoV-2. Front Microbiol. 2022 Mar 24;13:854209. doi: 10.3389/fmicb.2022.854209.”
- Table 1 has a section called Syndromic tests and yet this term has not been introduced in the introduction, nor has it been discussed in the Discussion section.
R: We introduced the “syndromic” term in the Introduction section. Additionally, syndromic tests were discussed in the Section 2, where they were listed. We added the following reference “Cassidy H, van Genne M, Lizarazo-Forero E, Niesters HGM, Gard L. Evaluation of the QIAstat-Dx RP2.0 and the BioFire FilmArray RP2.1 for the Rapid Detection of Respiratory Pathogens Including SARS-CoV-2. Front Microbiol. 2022 Mar 24;13:854209. doi: 10.3389/fmicb.2022.854209.”
- In Table 1, it will help if the specimen used is added, eg blood, saliva, nasal swabs etc
R: In Table 1 we added the “specimen” column to specify sample used by the reported molecular or serological assays.
- Is it possible to link the effect of Covid 19 on gut microbiota to wastewater epidemiology of the disease? In fact there are a few papers on this, such as Paterson, B.J. and Durrheim, D.N., 2022. Wastewater surveillance: an effective and adaptable surveillance tool in settings with a low prevalence of COVID-19. The Lancet Planetary Health, 6(2), pp.e87-e88. In fact Barua et al 2022 (Tracking the temporal variation of COVID-19 surges through wastewater-based epidemiology during the peak of the pandemic: A six-month long study in Charlotte, North Carolina. Science of the Total Environment, 814, p.152503) reported that SARS-CoV-2 in wastewater had 5–12 days lead time than clinical COVID-19 cases. Seeing the paper is on “microbiological and clinical findings of SARS-CoV-2 infection, perhaps that needs to be discussed too.
R: it is a very interesting observation. However, we don’t have a specific expertise on this specific topic, then we decided to not include it. We hope that you could understand our point of view.
- What do the names in lines 459-563 represent?
R: we corrected it.
Round 2
Reviewer 1 Report (New Reviewer)
I accept the authors explanations
Reviewer 2 Report (New Reviewer)
The authors addressed all of my comments, hence I recommend the publication of the current version.
Reviewer 3 Report (New Reviewer)
The authors have addressed most of my pertinent queries
This manuscript is a resubmission of an earlier submission. The following is a list of the peer review reports and author responses from that submission.
Round 1
Reviewer 1 Report
The manuscript by Russo et al put together information from papers mostly published in the last two years on early microbiological and clinical diagnosis of hospitalized patients with suspected COVID-19. They also described extrapulmonary manifestations and the role of gut microbiota in COVID-19.
This review is a timely summary as well as useful guidance in controlling the ongoing pandemic and improving the management of COVID-19 patients.
I only have a few questions and suggestions:
How many papers were included? Any information on the geographical and age distribution of the hospitalized patients from the literature?
Minor:
Line 39: “have not”, remove “not”.
Line 70: “routinely” to “routine”.
Table 1: “Molecularassay” to “Molecular assay”.
Table 1: Add the reference number for each reference.
Line 98: “are” to “is”.
Line 104, 114, 117, 148, 249: keep the reference numbering consistent.
Line 105: “show” to “shows”.
Line 121: “typically” to “typical”.
Line 163: provide reference.
Line 281-283 is the repeat of line 113-115. Consider rephrasing it.
Reviewer 2 Report
In this narrative review, Dr. Russo and colleagues attempted to discuss the recent update on microbiological and clinical management of SARS-CoV-2 infection after 2 years of pandemic. However, there are some fundamental issues with this manuscript that demand significant improvement:
o) This manuscript lacks of focus and I found it difficult to follow the story. In the introduction, the authors aimed to describe clinical and laboratory characteristics of hospitalized patients with suspected COVID-19. However, it is completely different than that mentioned in the title. I would suggest to restructure everything to make it clearer.
o) The abstract is also not clear and redundant. I can only extract a few points from it and the rest are just circling around the same issue. A good abstract should be able to summarize what is in the manuscript and inform the readers what to expect from the article and clearly those aspects were not fulfilled.
o) Since this is a narrative review, no "methods" section is allowed. If the authors want to change it to a systematic review, please conduct the study based on PRISMA guideline and register it in PROSPERO. Otherwise, please follow the common structure of a narrative review.
o) I think the authors need to improve the writing of this manuscript, including the English grammar. I found that the authors too often use "as the matter of fact" which does not have any meaning. Please write everything concisely and succinctly. This would help the readers to focus more on the contents and to avoid confusion. Consider asking for a help from a professional scientific writer or a Native English scientist.
o) "As a matter of fact, COVID-19 may show no peculiar signs and symptoms that may differentiate it from other infective or non-infective diseases; then, early recognition and prompt management are crucial to improve survival." If there is no specific signs or symptoms, how could we early diagnose COVID-19? What about anosmia and ageusia? Are they not specific for COVID-19 if accompanied with fever and (upper/lower) respiratory symptoms?
o) "Because of the pandemic caused by severe acute respiratory coronavirus-2 (SARS-CoV-2), the clinical understanding of the coronavirus disease-19 (COVID-19) in humans has deepened" This is an early example of the unclarity of the text. Because of the pandemic, clinical understanding of COVID-19 is improved? What does this mean? So, if there is no pandemic, clinical understanding of an infectious disease would not be improved?
o) Please use the citation format of MDPI.
o) Regarding section 3, which of those diagnostic tools are "new advances"? Please clarify. To me, they are already established even before COVID-19 pandemic and have been used to diagnose / characterize other diseases.
o) Regarding the sign and symptoms of COVID-19, it was clear that each virus variant has different disease pattern. For example, Omicron variant has upper respiratory preference. Please check this publication to better understand the context (https://doi.org/10.12688/f1000research.110647.1). I think it is better to separate this section based on the available variants. The authors can focus on the VOC only, since they are the ones causing health issues in the community.
o) Regarding the proposed management of COVID-19 in Figure 1, please clarify whether this protocol has been implemented in the authors' clinics / hospitals? Otherwise, the authors need to elaborate and justify the scientific backgrounds of those steps. Why should we follow those steps? What are the advantages of following those steps? Has it been tested in a trial?
o) This statement regarding corticosteroid therapy in COVID-19 "Data in literature showed that the use of dexamethasone resulted in lower 28-day mortality, especially in patients receiving invasive mechanical ventilation" can be improved by including some information from this publication (PMID: 34321903).
o) I think section 5 (extrapulmonary manifestation) should be incorporated in section 4 and discussed more extensively. Some patients might not have pathognomonic / clear respiratory issue but hypercoagulability can occur, causing thrombosis and other extrapulmonary damages (PMID: 34642600). It would also be clearer if the authors make a table including all the respiratory and non-respiratory symptoms and signs of COVID-19, divided by the variants or strains.
o) The authors could also extend the table by adding the relevant laboratory parameters that can be used to diagnose or predict the prognosis of particular strain or variant of COVID-19.
Reviewer 3 Report
Article: New advances in microbiological and clinical management of SARS-CoV-2 infection after 2 years of pandemic: a narrative review
The authors reviewed the “new” advances in the microbiological and clinical management of SARS-CoV-2 infection after 2 years of the pandemic, as described in the title. However, the authors only describe the techniques associated with the microbiology screening.
All the parts of the manuscript have important errors.
For example:
(Objective: The main objective of this review is to describe the clinical and laboratory characteristics of hospitalized patients with suspected COVID-19) – However, no information about the clinical and laboratory characteristics of hospitalized patients with suspected COVID-19 was inputted in the text.
Methods and results
The study included “Case series, controlled and uncontrolled studies, and meta-analyze”. In my opinion, how do the authors include meta-analyses in a review? Also, the authors did not cite the study type in the tables and figures.
Table 1 deserves to have a legend. In addition, the authors should include more information about each study, such as the conclusion of each study and the main findings.
Figure 1. The proposed management was based on the review done by the authors. In my opinion, the authors should cite references to blur the study. In addition, there are some errors in the figure, and the authors should include a legend.
Conclusions
The authors should delete the references from the text.